# Harvesting and amplifying gene cassettes confers cross-resistance to critically important antibiotics

**Punyawee Dulyayangkul[1,2], Thomas Beavis[1¤], Winnie W. Y. Lee[1], Robbie Ardagh[1], Frances Edwards[1,3], Fergus Hamilton[3], Ian Head[1,4], Kate J. Heesom[5], Oliver Mounsey[1], Marek Murarik[1], Peechanika Pinweha[1], Carlos Reding[1], Naphat Satapoomin[1], John M. Shaw[1], Yuiko Takebayashi[1], Catherine L. Tooke[1], James Spencer[1], Philip B. Williams[1,6], Matthew B. Avison[1] \***

1 School of Cellular & Molecular Medicine, University of Bristol, Bristol, United Kingdom, 2 Laboratory of Biotechnology, Chulabhorn Research Institute, Bangkok, Thailand, 3 North Bristol NHS Trust, Bristol, United Kingdom, 4 Somerset NHS Foundation Trust, Taunton, United Kingdom, 5 Bristol University Proteomics Facility, University of Bristol, Bristol, United Kingdom, 6 University Hospitals Bristol and Weston NHS Foundation Trust, Bristol, United Kingdom

¤ Current Address: European Molecular Biology Laboratory, Heidelberg, Germany
\* bimba@bristol.ac.uk

**Data Availability Statement:** All genome sequence data have been deposited onto the European Nucleotide Sequence Archive (ENA) under accession number PRJEB73531.

## Abstract

Amikacin and piperacillin/tazobactam are frequent antibiotic choices to treat bloodstream infection, which is commonly fatal and most often caused by bacteria from the family *Enterobacterales*. Here we show that two gene cassettes located side-by-side in and ancestral integron similar to In*37* have been "harvested" by insertion sequence IS*26* as a transposon that is widely disseminated among the *Enterobacterales*. This transposon encodes the enzymes AAC(6')-Ib-cr and OXA-1, reported, respectively, as amikacin and piperacillin/tazobactam resistance mechanisms. However, by studying bloodstream infection isolates from 769 patients from three hospitals serving a population of 1.2 million people in South West England, we show that increased enzyme production due to mutation in an IS*26*/In*37*-derived hybrid promoter or, more commonly, increased transposon copy number is required to simultaneously remove these two key therapeutic options; in many cases leaving only the last-resort antibiotic, meropenem. These findings may help improve the accuracy of predicting piperacillin/tazobactam treatment failure, allowing stratification of patients to receive meropenem or piperacillin/tazobactam, which may improve outcome and slow the emergence of meropenem resistance.

## Author summary

Piperacillin/tazobactam and amikacin are common antibiotic choices for treating bloodstream infections, which are potentially lethal and predominantly caused by the bacterium *Escherichia coli*. By studying a large collection of bloodstream *E. coli* we show that dual resistance to piperacillin/tazobactam and amikacin is unexpectedly common. We explain

**Funding:** This work was funded by Medical Research Council grants MR/T005408/1 (to P.B.W. and M.B.A), MR/N013646/1 (to M.B.A. and K.J.H.) and MR/S004769/1 (to M.B.A) and by Natural Environment Research Council grant NE/N01961X/1 (to M.B.A.), by grant 82459 (to M.B.A.) from the Welsh Government Rural Communities - Rural Development Programme 2014-2020 supported by the European Union and the Welsh Government (Llywodraeth Cymru). W.W.Y.L. received a scholarship from the Medical Research Foundation National PhD Training Program in Antimicrobial Resistance Research (MRF-145-0004-TPG-AVISO). Clinical training fellowships were funded by the Wellcome Trust (to F.E. and F.H) and National Institute for Health Research (to I.H.). P.P. was supported by a Royal Thai Government scholarship. N.S. was supported by a postgraduate scholarship from the University of Bristol. The funders had no role in study design, data collection and analysis, decision to publish, or preparation of the manuscript. No author received a salary directly from any funder.

**Competing interests:** The authors have declared that no competing interests exist.

this by showing that two resistance genes are circulating widely as part of a transposon. We show that genetic changes in the transposon, most commonly an increase in the number of copies of the transposon in the genome, are required to confer co-resistance to both antibiotics. Hence we have identified an important, emerging, mobile genetic element with the potential to compromise therapy by two commonly used antibiotics for bloodstream infection. We also show how this knowledge might be used to improve the sensitivity of tests for piperacillin/tazobactam resistance by integrating the results of amikacin resistance testing, reducing treatment failure and sparing last resort antibiotics.

## Introduction

Sepsis is a life-threatening syndrome frequently resulting from bacterial bloodstream infection (BSI) that caused around 20% of deaths worldwide in 2017 [1]. Antibiotic resistance increases mortality rates, and in 2019, approximately 1.5 million deaths globally were associated with antibiotic resistant (ABR) BSI, of which around 400,000 deaths could be directly attributed to ABR bacteria [2]. In high income countries, BSI is primarily opportunistic, occurring when bacteria–most commonly of the species *Escherichia coli*–enter the blood through the urinary or gastrointestinal tracts, wounds, indwelling medical devices and from other sources [3].

Rapid treatment of BSI with an effective antibiotic greatly improves outcome, so when BSI is suspected, patients are usually prescribed antibiotics before microbiological confirmation of the causative organism and its antibiotic susceptibility, which can take several days [3]. The antibiotics of choice vary based on local epidemiology, but tend to be β-lactams, frequently the penicillin/penicillinase inhibitor combination piperacillin/tazobactam, or a third-generation cephalosporin (3GC) [4]. β-lactam resistance in *E. coli* is primarily due to β-lactamase enzymes, which include CTX-M-15 (3GC resistance) [5] and OXA-1 (piperacillin/tazobactam resistance) [6]. Where β-lactams cannot be used, and in regions where carbapenem use is being minimised as a matter of policy, aminoglycoside antibiotics, either amikacin or gentamicin, are the usual alternative [4]. In *E. coli*, resistance to these agents is predominantly due to aminoglycoside acetyltransferase (AAC) enzymes of the AAC(6')-I (amikacin resistance) or AAC(3)-II (gentamicin resistance) types. Both inactivate their substrate by transferring an acetyl group from Acetyl-CoA to a specific position on the aminoglycoside scaffold [7].

OXA-1, AAC(6')-I and AAC(3)-II, are encoded by gene cassettes [8–10]. These are promoter-less genes flanked by specific sequences used to insert them into integrons—gene cassette capture structures commonly found in bacterial genomes [11]. A wide variety of gene cassettes exists, making integrons highly variable. Integrons provide an upstream promoter, driving expression of the gene cassettes present, and encode an integrase enzyme capable of capturing additional cassettes [11]. Importantly, integrons are not themselves able to move between sites on DNA, e.g. from a plasmid onto the chromosome. However, integrons can be moved as part of a composite transposon, where they are flanked by two copies of the same insertion sequence (IS), which encodes a transposase enzyme that can move the transposon, potentially resulting in transposon replication [11].

It has been noted that piperacillin/tazobactam resistance and amikacin resistance are phenotypically linked in some *E. coli* populations, particularly those that are also 3GC resistant [12]. This is important given the primacy of these agents for BSI therapy and the serious consequences of treatment failure. Our present study was initiated based on analysis of antibiotic susceptibility patterns among BSI *E. coli* cultured at a regional diagnostic microbiology lab serving healthcare infrastructure for a population of approximately 1.2 million people in South

West England. We noted that amikacin resistance (classed as being intermediate or resistant based on standard EUCAST clinical breakpoints [13]) in BSI *E. coli* appears to be focussed among 3GC resistant isolates that are also piperacillin/tazobactam resistant. Here we sought the genetic reason for this association, and identified a process where contiguous pairs of gene cassettes from circulating integrons can be mobilised by an insertion sequence as part of a composite transposon. In this case, we find the transposon responsible to be widely distributed among bacteria of the family *Enterobacterales*, and can have increased copy number, causing co-resistance to critically important therapies for BSI in many healthcare settings.

## Results

### Carriage of *aac(6')-Ib-cr* does not always confer amikacin resistance in *E. coli*

In 2020, in our study region, the amikacin resistance rate, based on routine diagnostic microbiology among deduplicated isolates from 838 *E. coli* culture-positive BSIs (cultured from 824 individual blood samples), was 5.7%. Among 3GC-resistant *E. coli* isolates (defined using cefotaxime as the indicator) which caused 83 BSIs, the amikacin resistance rate was 36.1% ($\chi^2$ = 157.8, p < 0.00001). We were able to retrospectively recover 71 of these 3GC-resistant isolates from 2020 for whole genome sequencing (WGS) and supplemented this dataset with 69 additional 3GC-resistant BSI *E. coli* collected in the same region during 2018. Analysis of WGS data revealed that $bla_{\text{CTX-M-15}}$ is the dominant cause of 3GC resistance (107/140, 76% of isolates), with 67/107 (63%) of $bla_{\text{CTX-M-15}}$ positive isolates also carrying *aac(6')-Ib-cr*, which was absent from all 33 $bla_{\text{CTX-M-15}}$ negative 3GC-resistant isolates (S1 Table). Overall, 48% of sequenced 3GC-resistant isolates carried *aac(6')-Ib-cr*, which encodes a variant of AAC(6')-I [9,14] reported as an amikacin resistance mechanism [15]. This provides a putative mechanistic explanation for the observed association between 3GC resistance and amikacin resistance among BSI *E. coli*. However, 14/67 (21%) of the *aac(6')-Ib-cr*-positive 3GC-resistant isolates were amikacin susceptible (S1 Table). This led us to question whether carriage of *aac(6')-Ib-cr* can universally confer amikacin resistance in *E. coli*, as defined by diagnostic laboratory susceptibility testing at the time these study isolates were recovered.

We next sequenced 629 3GC-susceptible BSI *E. coli* from the same region, also collected in 2020. Eight were found to carry *aac(6')-Ib-cr*, and of these, two (25%) were amikacin susceptible; a similar susceptibility rate to the 21% seen among 3GC-resistant *aac(6')-Ib-cr*-positive isolates. One of the total of 16 sequenced amikacin-susceptible *aac(6')-Ib-cr*-positive isolates identified here had a deletion spanning the initiation codon of *aac(6')-Ib-cr*, explaining why this isolate is amikacin susceptible (S1 Table), but in the remainder, no mutation in *aac(6')-Ib-cr* was found. Hence, in total, 15/75 (20%) of sequenced *aac(6')-Ib-cr* positive bloodstream *E. coli* were amikacin susceptible without any explanation based on sequence. We took this as evidence that *aac(6')-Ib-cr* is not always a determinant of amikacin resistance.

### AAC(6')-Ib-cr is a weakened amikacin resistance determinant

In order to confirm that AAC(6')-Ib-cr can acetylate amikacin, *aac(6')-Ib-cr* was overexpressed in *E. coli* and the recombinant enzyme purified to homogeneity. *In vitro* assays demonstrated activity of AAC(6')-Ib-cr against amikacin, albeit to a lower level than towards the structurally related aminoglycoside antibiotic kanamycin (Fig 1).

Despite this demonstration of *in vitro* activity towards amikacin, recombinant *E. coli* carrying *aac(6')-Ib-cr* ligated into a low copy number vector [16] remained amikacin susceptible based on broth microdilution MIC assays, although the recombinant strain was kanamycin

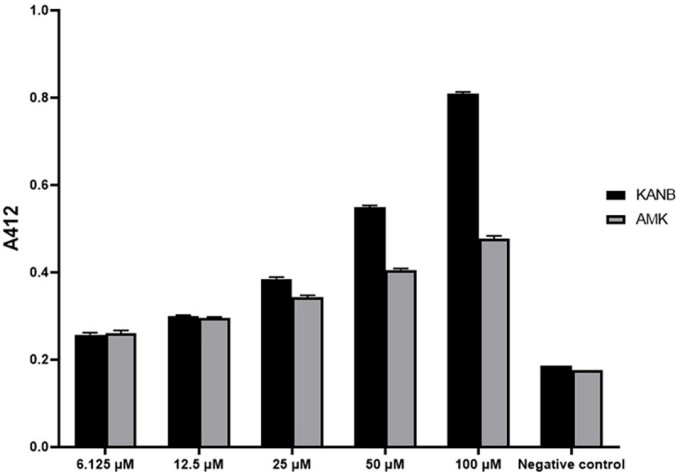

**Fig 1. Enzymatic assay illustrating ability of AAC(6')-Ib-cr to modify both kanamycin B and amikacin.** Increasing concentrations of aminoglycoside substrates were incubated with 100 nM of purified enzyme and 0.1 mM acetyl-CoA for 1 h to allow enzymatic reaction to take place. Enzymatic activity was then measured by adding 2 mM DTNB and monitoring the absorbance at 412 nm of the by-product from an acetylation reaction. Enzymatic activity against kanamycin B as a known substrate, and amikacin as a suspected substrate, increased proportional to substrate concentration and above negative control. This suggests AAC(6')-Ib-cr has the ability to modify amikacin. Negative controls contain no acetyl-coA and thus no measurable acetylation.

resistant, and the amikacin MIC increased four-fold over vector-only control (Table 1). Together with our *in vitro* results, these data indicate that AAC(6')-Ib-cr is active towards amikacin, but, under these conditions, at a level that may not always be sufficient to cause amikacin resistance.

AAC(6')-Ib-cr was discovered because it has two amino acid changes relative to the parent enzyme AAC(6')-Ib, which collectively expand substrate spectrum to include some fluoroquinolones including ciprofloxacin [9]. Site-directed mutagenesis reverting *aac(6')-Ib-cr* to its parent *aac(6')-Ib* in our recombinant *E. coli* model system resulted in a stepwise reduction in ciprofloxacin MIC, as expected, with the double, and one single, reversion mutation conferring amikacin resistance (Table 1). We conclude, therefore, that this increase in substrate spectrum of AAC(6')-Ib-cr to encompass fluoroquinolones comes at the cost of reduced activity towards amikacin. This led to a hypothesis to explain why 20% of sequenced BSI *E. coli* carrying *aac (6')-Ib-cr* in our region were amikacin susceptible (S1 Table): elevated expression of this gene may be necessary to confer amikacin resistance. Importantly, *aac(6')-Ib-cr* was the only aminoglycoside resistance gene in 32/38 of sequenced amikacin-resistant BSI *E. coli* (S1 Table).

**Table 1. Broth microdilution MICs of ciprofloxacin, kanamycin and amikacin against *E. coli* carrying *aac(6')-Ib-cr* and variants.**

| Strain | MIC (mg/L) of | | |
|---|---|---|---|
| | Ciprofloxacin | Kanamycin | Amikacin |
| *E. coli* XL10 gold (plasmid only) | 0.0625 (S) | 4 (S) | 2 (S) |
| *E. coli* XL10 gold (*aac(6')-Ib-cr*) | 0.25 (S) | >128 (R) | 8 (S) |
| *E. coli* XL10 gold (*aac(6')-Ib-cr* Y179D) | 0.0625 (S) | >128 (R) | 8 (S) |
| *E. coli* XL10 gold (*aac(6')-Ib-cr* R102W) | 0.125 (S) | >128 (R) | 16 (R) |
| *E. coli* XL10 gold (*aac(6')-Ib-cr* R102W, Y179D) | 0.0625 (S) | >128 (R) | 64 (R) |

S, susceptible; R, resistant

Hence, if our hypothesis is correct, these data identify *aac(6')-Ib-cr* over-expression as the dominant cause of amikacin resistance in BSI *E. coli* in our region.

## Capture of *aac(6')-Ib-cr* by IS*26* allows amikacin resistance due to promoter activation or increased gene copy-number

In all sequenced *aac(6')-Ib-cr*-positive BSI *E. coli* in our survey, whether amikacin susceptible or not, a copy of the insertion sequence IS*26* was found upstream of *aac(6')-Ib-cr*, and always located in the same position. The *aac(6')-Ib-cr* gene cassette appears, therefore, to have been mobilised from its original location within an integron. The previously reported integron In*37* [17] carries *aac(6')-Ib-cr*, the β-lactamase gene *bla*$_{OXA-1}$ and the chloramphenicol acetyltransferase gene *catB* as its first three gene cassettes (Fig 2A). In 66/74 *aac(6')-Ib-cr* positive BSI *E. coli* sequenced here, these three gene cassettes are located downstream of IS*26* –suggesting that In*37* is their ancestral origin–but *catB* is truncated at an identical position in all by the insertion of a second copy of IS*26*, creating a composite transposon (Fig 2B). In the remaining eight isolates, this second IS*26* copy is in a variety of positions, either truncating or entirely removing *bla*$_{OXA-1}$ (S1 Table).

IS*26* is known to provide an outward-facing -35 promoter element (TTGCAA) having 4/6 matches to the consensus *E. coli* -35 promoter element (TTGACA) [18]. In the specific insertion seen here, this IS*26*-derived -35 element is found optimally positioned 17 nucleotides upstream of a CATAAT sequence that is normally present 5' proximal to *aac(6')-Ib-cr* in the integron In*37* and has 5/6 matches to the consensus *E. coli* -10 promoter element (TATAAT) (Fig 2C). We confirmed that this is an active promoter using 5'RACE (Fig 2C). None of the amikacin-susceptible *aac(6')-Ib-cr* positive BSI *E. coli* isolates sequenced here had a mutation in this promoter or anywhere within IS*26* or upstream of *aac(6')-Ib-cr* that might explain

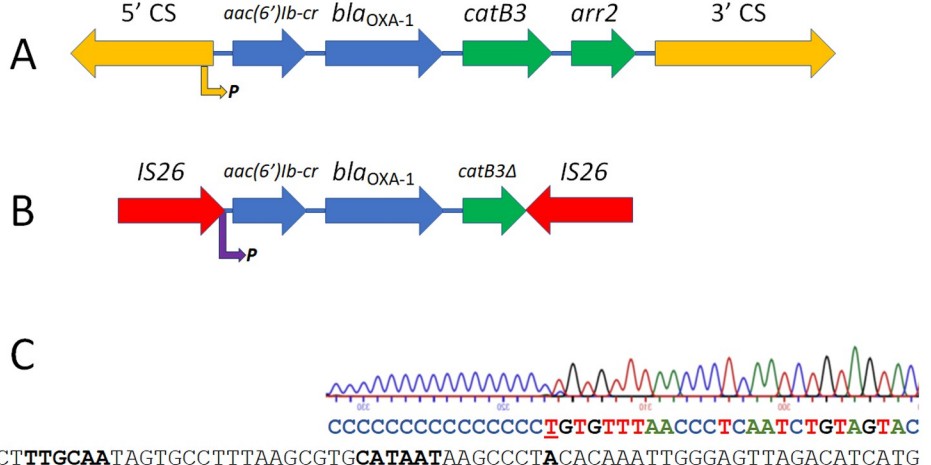

**Fig 2. Diagrammatic representation of In*37* and the derivative transposon and definition of the IS*26*/In*37* hybrid promoter.** The genes making up integron In*37* (A) and the derived transposon (B) are presented to scale. In*37* is a class 1 integron, with 5' and 3' conserved sequences (CS; orange), where the 5'CS encodes the integrase enzyme and an outward facing promoter that drives expression of the gene cassettes as a single transcript (orange arrow to P). In (B) the insertion sites of two copies of IS*26* are noted (red), where this generates a composite transposon including a hybrid IS*26*/In*37* promoter sequence (purple arrow to P), and mobilises *aac(6')-Ib-cr* and *bla*$_{OXA-1}$ (blue) but excludes the two distal gene cassettes from In*37* (green) with the *catB3* gene being truncated in the process (noted as *catB3Δ*). In (C) we confirm the location of the *aac(6')-Ib-cr/bla*$_{OXA-1}$ operon transcriptional start site using 5'RACE, and present the sequence trace resulting from this experiment, showing poly-C extension and pinpointing the transcriptional start site in complementary sequence relative to the the -10 and -35 promoter elements (bold and underlined) in the IS*26*/In*37* hybrid promoter.

susceptbility Notably, however, two amikacin-resistant, *aac(6')-Ib-cr* positive isolates had an identical C-T transition, strengthening this putative -10 promoter sequence and generating a perfect consensus (S1 Table). Accordingly, we hypothesised that these modifications may increase *aac(6')-Ib-cr* expression, potentially explaining amikacin resistance in these two isolates. LC-MS/MS proteomics confirmed that, when normalised to gene copy number and ribosomal protein abundance, this mutation increases AAC(6')-Ib-cr production approximately 4-fold (Fig 3A), confirming that this IS*26*/In*37* hybrid promoter can be made stronger by mutation.

IS*26*-mediated composite transposons are known to have the potential to duplicate, increasing copy number and therefore gene expression [18]; plasmid-located transposon copy

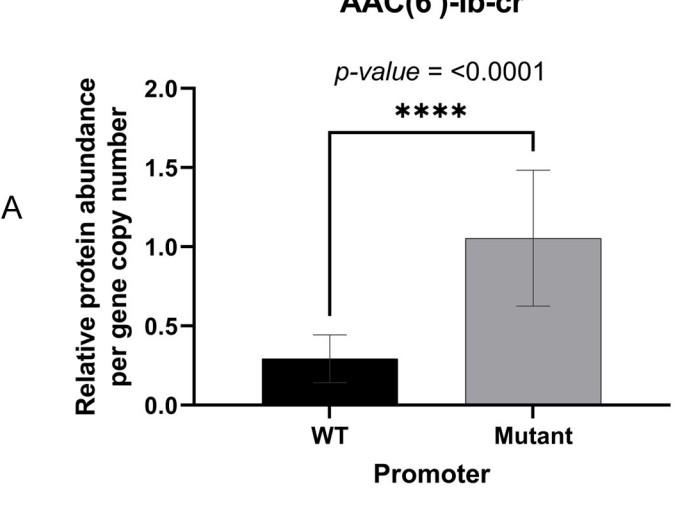

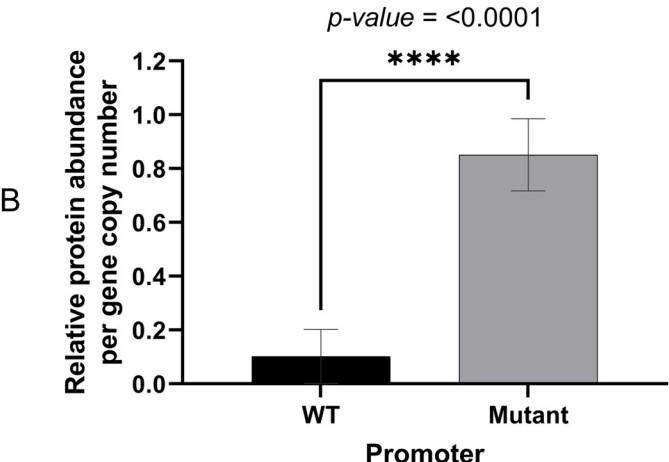

**Fig 3. Relative AAC(6')-lb-cr and OXA-1 protein abundance in IS*26*/In*37* hybrid transposon promoter mutants versus wild-type.** Bar charts show a relative protein abundance per gene copy number of (A) AAC(6')-lb-cr and (B) OXA-1 with wild-type (WT) promoter and mutated (mutant) promoter, which has a closer match to the -10 *E. coli* promoter consensus, (TATAAT versus CATAAT as show in Fig 2C). **** indicates *p*-value ≤ 0.0001 by unpaired t-test.

number is also dictated by the copy number of the plasmid carrying it. Of the 72 sequenced *aac(6')-Ib-cr* positive BSI *E. coli* having a wild-type IS*26*/In*37* hybrid promoter, 15 were amikacin susceptible and 57 were amikacin resistant. We hypothesised that higher *aac(6')-Ib-cr* copy number would mean elevated gene expression, and that in the absence of promoter mutations this is the mechanism by which amikacin resistance occurs.

To test this hypothesis, the gene copy number for *aac(6')-Ib-cr* was determined based on read density analysis from WGS data and isolates with differing copy numbers were selected for LC-MS/MS proteomics. A linear association was observed between AAC(6')-Ib-cr protein abundance and gene copy number ($R^2$ = 0.76, Fig 4A). The single outlier is an isolate with exceptionally high *aac(6')-Ib-cr* copy number, where protein abundance appears to be saturated, either in our assay, or in the cell (Fig 4B).

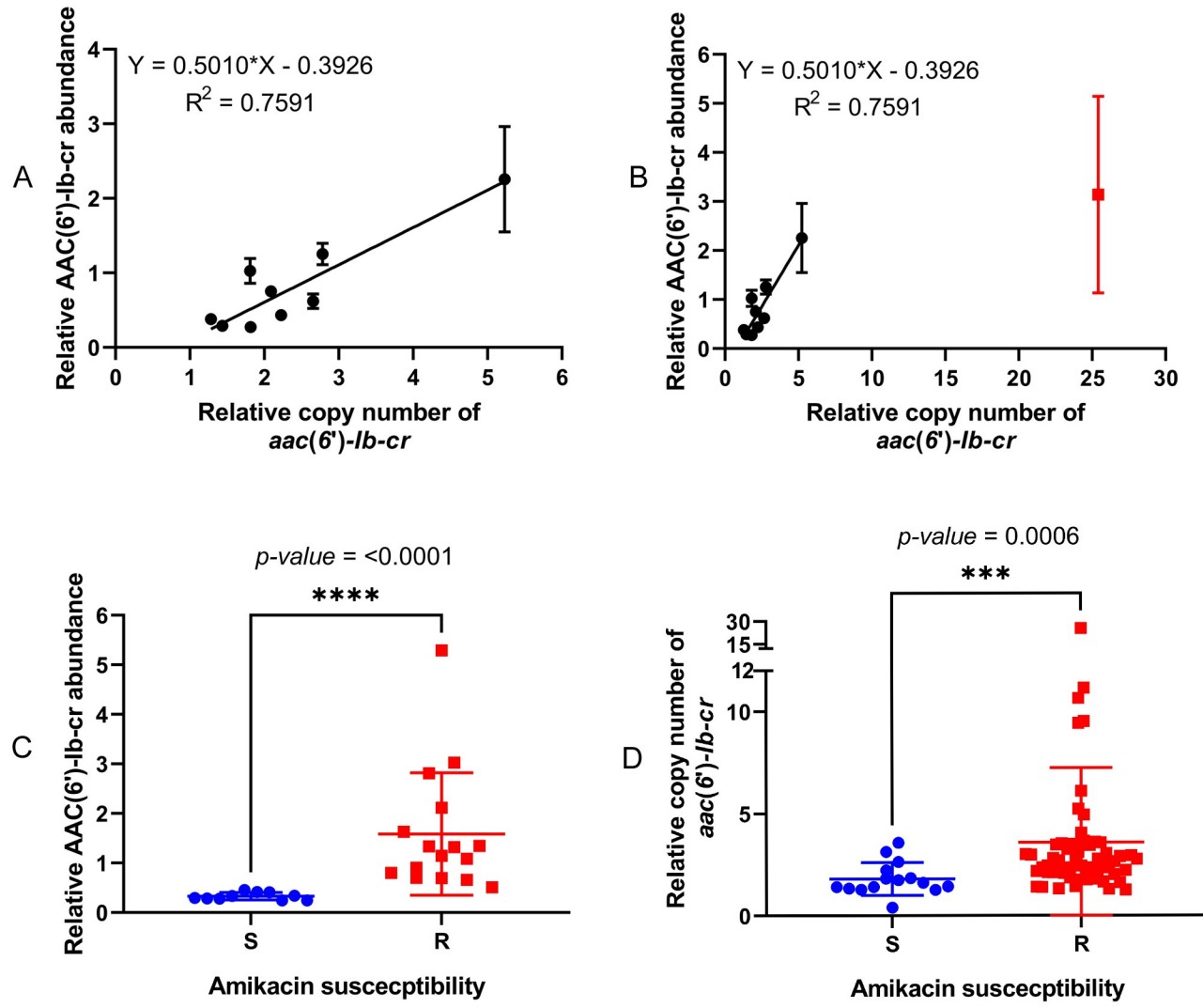

**Fig 4. Association between AAC(6')-lb-cr abundance, *acc(6')-lb-cr* copy number and amikacin susceptibility.** (A) A linear correlation of AAC(6')-lb-cr abundance and *acc(6')-lb-cr* copy number, an outlier was shown in red (B). Association of amikacin susceptibility phenotype with (C) relative AAC(6')-lb-cr abundance and (D) relative *aac(6')-lb-cr* copy number. S, susceptible; R, resistant; **** indicates *p*-value ≤ 0.0001 and *** indicates *p*-value ≤ 0.001 by Mann-Whitney test.

In further support of our hypothesis, we identified a strong association between AAC(6')-Ib-cr abundance in cells and amikacin susceptibility/resistance status in those isolates subjected to proteomic analysis ([Fig 4C]). Finally, when considering all 72 *aac(6')-Ib-cr* positive BSI isolates having a wild-type IS*26*/In*37* hybrid promoter, it is apparent that our measure of gene copy number based on average read density in WGS data provides a strong predictor of amikacin susceptibility/resistance status ([Fig 4D]).

### Co-ordinately increased *bla*<sub>OXA-1</sub> expression alongside aac(6')-Ib-cr expression confers piperacillin/tazobactam resistance

We have already observed that 66 *aac(6')-Ib-cr* positive BSI isolates sequenced here carried the same IS*26*-mediated transposon, having fully intact *aac(6')-Ib-cr* and *bla*<sub>OXA-1</sub> genes ([Fig 2B] and [S1 Table]). Among these isolates there was a linear correlation between the gene copy numbers of *aac(6')-Ib-cr* and *bla*<sub>OXA-1</sub>, indicative of a situation where the entire transposon has a higher or lower copy number in different isolates ([Fig 5A]). As with AAC(6')-Ib-cr, the

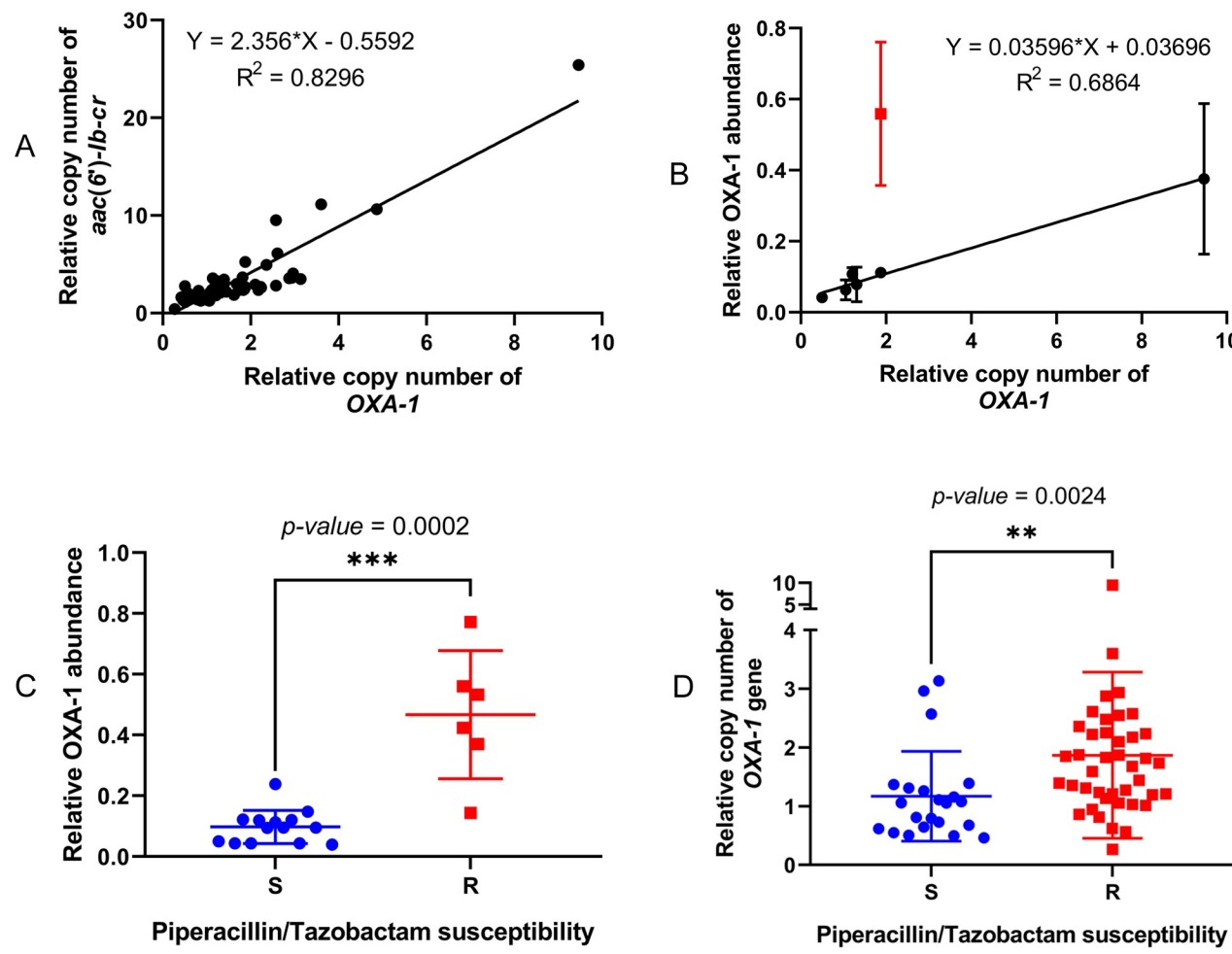

**Fig 5. Associations between *bla*<sub>OXA-1</sub> and *aac(6')-Ib-cr* copy number, OXA-1 abundance and Piperacillin/Tazobactam susceptibility.** A linear correlation of relative OXA-1 and (A) *acc(6')-Ib-cr* or (B) relative OXA-1 abundance where the outliner shows in red. Association of Piperacillin/Tazobactam susceptibility phenotype with (C) relative OXA-1 abundance and (D) relative OXA-1 gene copy number. S, susceptible; R, resistant; *** indicates *p*-value ≤ 0.001 and ** indicates *p*-value ≤ 0.01 by Mann-Whitney test.

activating mutation in the hybrid IS$26$/In$37$ promoter increased OXA-1 production (Fig 3B), even though $bla_{OXA-1}$ is the second gene cassette, and so further away from the promoter (Fig 2B). A strong correlation was observed between $bla_{OXA-1}$ gene copy number and OXA-1 production, as measured by proteomics–though again there was one unexpected outlier (Fig 5B).

OXA-1 production has previously been associated with piperacillin/tazobactam resistance [19]. We identified, however, that as is the case for the association between AAC(6')-Ib-cr production level and amikacin resistance, increased production of OXA-1 is necessary for piperacillin/tazobactam resistance, based on disc susceptibility testing, among BSI *E. coli* carrying the IS$26$-mediated composite transposon that includes both *aac(6')-Ib-cr* and $bla_{OXA-1}$ (Fig 5C).

Piperacillin/tazobactam MIC is affected by multiple factors in *E. coli*, e.g., envelope permeability and expression of β-lactamase enzymes other than OXA-1 [19–26]. After excluding isolates with additional β-lactamase genes, we found that $bla_{OXA-1}$ copy number within the IS$26$-mediated composite transposon (Fig 2B) was associated with piperacillin/tazobactam susceptibility/resistance status (Fig 5D), though this association was not as strong as that between *aac(6')-Ib-cr* copy number and amikacin susceptibility/resistance status ($p = 0.002$ vs $p = 0.0006$; Figs 5D and 4D), likely because piperacillin/tazobactam resistance is multi-factorial, whereas amikacin resistance in *E. coli* is almost exclusively caused by an aminoglycoside modifying enzyme [15].

## Discussion

We have previously reported linkage between $bla_{CTX-M-15}$, $bla_{OXA-1}$ and *aac(6')-Ib-cr* carriage in urinary *E. coli* from our region in South West England [27], here confirmed among BSI isolates. Recently, a genomic survey of urinary *E. coli* isolated in the U.S. also reported linkage between $bla_{CTX-M-15}$, $bla_{OXA-1}$ and *aac(6')-Ib-cr* [12]. In that study, Jackson *et al.*, noted that *aac(6')-Ib-cr* carriage was not universally associated with amikacin resistance, and speculated that *aac(6')-Ib-cr*-positive amikacin-resistant isolates carried additional amikacin resistance mechanisms, not present in the *aac(6')-Ib-cr*-positive amikacin susceptible isolates, but could not identify these additional mechanisms from WGS data [12]. Here we show that increased *aac(6')-Ib-cr* copy number provides the likely explanation for this switch from amikacin susceptibility to resistance in our region. To test whether this explanation could also be applied to the U.S. study [12], we consulted a list of *aac(6')-Ib-cr*-positive isolates provided by Jackson *et al.*, alongside their amikacin susceptibility status. Our analysis of read density from raw sequence data deposited by the authors (BioProject PRJNA891712) showed that the *aac(6')-Ib-cr* copy number was 4.82 +/- 2.01 (mean +/- standard error of the mean [SEM], n = 14) for isolates defined as amikacin non-susceptible (resistant or intermediate based on 2020 CLSI criteria [12]) and 1.52 +/- 0.05 (n = 150) for amikacin susceptible isolates. This difference is significant (t-test, $p<0.0001$).

IS$26$ is renowned as a pathway to composite transposon formation and mobilisation [18], first coming to prominence as the mechanism by which $bla_{SHV}$ was mobilised from the *Klebsiella pneumoniae* chromosome prior to mutation into a group of extended spectrum β-lactamase (ESBL) variants, which then spread globally as a 3GC resistance mechanism in the *Enterobacterales* [28]. Recently, IS$26$ was shown also to have captured $bla_{TEM-1}$, the most common acquired amoxicillin resistance mechanism in *E. coli*. As well as well as being frequently found on high copy number plasmids, the copy number of the resultant composite transposon can be amplified by IS$26$, increasing TEM-1 production and the MICs of various amino-penicillin/β-lactamase inhibitor combinations, including piperacillin/tazobactam [23–25]. Furthermore, in an *in vitro* mutant selection study using a model *E. coli* system and recombinant

plasmid, it has recently been reported that $bla_{OXA-1}$ copy number amplification was associated with evolution towards piperacillin/tazobactam resistance in the laboratory, and was driven by IS*26* [29].

Here we have identified another IS*26* mobilisation event with significant clinical implications, generating a 3699-nucleotide composite transposon (Fig 2B). The *aac(6')-Ib-cr* variant gene cassette was first found in a clinical isolate collected in 2003, and was shown to encode an enzyme capable of acetylating ciprofloxacin [9]. Whilst this gene cannot confer ciprofloxacin resistance alone [16], it does contribute to survival in the presence of the drug even in bacteria where ciprofloxacin resistance is caused by target site mutations [30].

Likely because this variant is relatively recently emerged, *aac(6')-Ib-cr* is found in only 17 INTEGRALL integron database entries out of a total of 336 integrons carrying *aac(6')-Ib-* variants [31]. Furthermore, in 11/17 integrons on this database, *aac(6')-Ib-cr* is the first gene cassette, being closest to the integron promoter (gene cassettes are promoter-less [11]) followed by $bla_{OXA-1}$ and *catB3* in a polycistronic structure first seen in the integron In*37* [17] (Fig 2A). The predominance of this arrangement appears to have provided the opportunity for IS*26* to mobilise two critically important resistance genes together, providing an upstream hybrid promoter and the potential for *aac(6')-Ib-cr* and $bla_{OXA-1}$ to be hyper-expressed together. This can be either due to hybrid promoter strengthening mutation (Fig 3) or gene copy number increase (Figs 4B and 5B). Both promoter mutation and increased copy number were seen in clinical isolates in this study, but increased copy number was more common. We have not defined whether increased copy number is due to mobilisation onto high copy number plasmids or amplification of the transposon via IS*26*. and this is likely to vary from isolate to isolate. The result is a phenotype compromising two therapies for BSI, piperacillin/tazobactam and amikacin, that are commonly used where the causative pathogen is already 3GC-resistant, or where clinicians wish to spare 3GC use for other reasons.

Using blastn, an identical 3699 nucleotide IS*26-aac(6')-Ib-cr-bla*$_{OXA-1}$-IS*26* composite transposon (Fig 2B) was detected at 100% identity and coverage in 714 of all extant NCBI database entries (10th June 2023), including 221 *E. coli* and 434 *Klebsiella* spp. entries. The first published report referring to one of these database entries was in 2010 [32]. This is extremely concerning because, whilst information in the database does not make it possible to calculate copy number for these strains, it is clear that the raw materials required for coordinated loss of piperacillin/tazobactam and amikacin from the therapeutic armamentarium are already widely disseminated. Indeed, to add evidence for this potential, we performed a secondary analysis of raw sequence read data deposited by Jackson *et al.* [12] and found that of 164 *aac(6')-Ib-cr* positive 3GC-resistant isolates where a copy number could be defined, 160 also carried $bla_{OXA-1}$ and the copy numbers of these two genes were highly correlated ($R^2 = 0.994$). Of 22 *aac(6')-Ib-cr*/$bla_{OXA-1}$ positive isolates defined as piperacillin/tazobactam non susceptible (intermediate or resistant based on CLSI breakpoints from 2020) by Jackson *et al.* [12] the average $bla_{OXA-1}$ and *aac(6')-Ib-cr* copy numbers were 3.13 +/- 1.29 and 3.49 +/- 1.53 (mean +/- SEM), respectively where for piperacillin/tazobactam susceptible isolates the copy numbers were 1.19 +/- 0.05 and 1.32 +/- 0.06 (n = 138), for $bla_{OXA-1}$ and *aac(6')-Ib-cr*, respectively. Both are statistically significant differences (t-test, $p<0.0001$ for $bla_{OXA-1}$ and $p = 0.0001$ for *aac(6')-Ib-cr*), confirming that the findings in our region, are more generally applicable.

There has been some interest in the utility of piperacillin/tazobactam and amikacin as combination therapy, and pharmacodynamics comparable to meropenem have been reported against 3GC-resistant *E. coli* carrying $bla_{CTX-M-15}$, $bla_{OXA-1}$ and *aac(6')-Ib-cr* [33]. It is entirely possible that the two agents could work additively (e.g., by piperacillin/tazobactam increasing amikacin penetration) but our work reported here suggests potential problems. We do not know the copy number maximum for the transposon identified here, or the resultant

piperacillin/tazobactam and amikacin MICs. Therefore, thorough testing of isolates with high transposon copy number is advised during future validation of this combination.

In isolates from our region resistant to piperacillin/tazobactam and amikacin, few therapeutic options remain: 3GCs, gentamicin and meropenem being the most widely used. Of 53 sequenced amikacin-resistant bloodstream isolates carrying the transposon identified here, 94% were 3GC-resistant and 34% were gentamicin-resistant. This compares with 3GC- and gentamicin-resistance rates of 9.7% and 8.9%, respectively, among all BSI isolates collected in our region in 2020. In contrast, 100% of the transposon-carrying isolates in our study were susceptible to meropenem, widely considered a last resort antibiotic.

This resonates with findings from the Australasian MERINO study that piperacillin/tazobactam cannot be considered non-inferior to meropenem against 3GC-resistant (predominantly CTX-M-producing) but piperacillin/tazobactam-susceptible (based on diagnostic laboratory tests) *E. coli* and *K. pneumoniae* BSI; and in consequence that use of piperacillin/tazobactam as a meropenem-sparing therapy could not be supported [34]. However, a more recent retrospective cohort study suggests that piperacillin/tazobactam is non-inferior to meropenem when BSI is caused by 3GC-resistant *E. coli* in geographical regions where $bla_{OXA-1}$ is uncommon [35]. One possible reason for this is that there can be considerable diagnostic uncertainty about piperacillin/tazobactam susceptibility in OXA-1-producing isolates, and particularly with the specific method used to test isolates in the MERINO trial [36], though it is clear from a number of clinical surveillance studies that carriage of $bla_{OXA-1}$ is associated with elevated piperacillin/tazobactam MIC [6,19,37]. Our work reported here, and our secondary analysis of the data generated by Jackson et al. [12] suggests that elevated $bla_{OXA-1}$ expression, rather than simply its presence, is associated with piperacillin/tazobactam resistance. It may be that this elevated expression is not detected by some or all currently available clinical tests for piperacillin/tazobactam susceptibility. It may be, also, that consideration of the amikacin susceptible/resistant phenotype in these isolates, which we have shown here is strongly associated with *aac(6')-Ib-cr*, and so $bla_{OXA-1}$, expression (Figs 4 and 5, and our secondary analysis, above) will be useful as a way to predict potential piperacillin/tazobactam treatment failure, even if clinical diagnostics indicate piperacillin/tazobactam susceptibility. Hence knowledge of amikacin susceptibility may be a way of stratifying patients to receive piperacillin/tazobactam or meropenem, improving treatment outcomes whilst sparing meropenem use until there is no other viable option. This will require validation, and it is important to note that should the breakpoint MIC defining amikacin resistance be reduced, this would reduce the sensitivity of using the amikacin resistance flag to help define piperacillin/tazobactam resistance. However, it would be possible for diagnostic labs to apply two "breakpoints" for amikacin, one to increase the sensitivity of calling piperacillin/tazobactam resistance, and one to call amikacin resistance.

## Materials and methods

### Isolates, antibiotic susceptibility testing and ethics statement

Isolates were collected from patients with BSI at three hospitals: Bristol Royal Infirmary, Southmead Hospital and the Royal United Hospital, Bath; collectively serving a population of approximately 1.2 million people in the South West of England. Blood samples were collected throughout 2018 and 2020 and processed for routine diagnostic microbiology at a single laboratory, Severn Pathology. Culture-positive samples were further processed by Gram stain, and bacterial species identified using MALDI-TOF mass spectrometry. Antibiotic susceptibility profiles were determined using disc diffusion directly from culture positive blood (piperacillin/tazobactam only, because preliminary analysis of BSI *E. coli* in our region using EUCAST

broth microdilution MIC testing [13] indicated that EUCAST disc testing is more sensitive than Vitek 2 at calling piperacillin/tazobactam resistance) or using Vitek 2 (Card AST-N350; ref. 421037) following initial culture on agar (all other agents). Susceptibility/resistance was defined using EUCAST breakpoints [13] at the time of isolation; where "intermediate" break-points were in use, intermediate isolates were defined as resistant. The result for cefotaxime was used to define 3GC susceptibility/resistance. Isolates were deduplicated by patient and antibiotic susceptibility profile using a >7 day re-sampling limit. This project is not part of a trial or wider clinical study requiring ethical review. There were no live study participants and only secondary data was used. All bacterial isolates came from routine diagnostic samples and no patient data was recorded or used.

## WGS and data analysis

Genomes were sequenced by MicrobesNG (Birmingham, UK) on a HiSeq 2500 instrument (Illumina, San Diego, CA, USA). Reads were trimmed using Trimmomatic v0.39 [38] and assembled into contigs using SPAdes [39] v3.13.0 (http://cab.spbu.ru/software/spades/). and contigs were annotated using Prokka v1.14.5 [40]. The presence of resistance genes was determined with ABRicate v1.0.1 using the ResFinder v2.1 database [41]. Gene copy number was determined using a bespoke pipeline, Hound [42], where raw reads were mapped against the assembled genome using the Burrows-Wheeler aligner BWA [43] and the resulting alignments were processed with SAMtools to extract read depth [44].

## Cloning and mutagenesis of *aac(6')-Ib-cr*

The *aac(6′)-Ib-cr* gene was synthesized (Eurofins Genomics) to include its native promoter from 5-'GAGCAAACGATCAATGCG to AAGCCTAGCCAATTTGCTAGG-3', as presented in Genbank accession number JF775514.1, and ligated into the pEX-K4 cloning vector (Euro-fins). The insert was subcloned into the low copy number plasmid pSU18 and recombinants used to transform *E. coli* XL10-Gold (Stratagene) to chloramphenicol resistance (30 mg/L). Site directed mutagenesis was performed using the Quick-Change Lightning kit (Agilent Technologies) and primers "R-W mutant" (5'-CCGTAACCACCCCAGATGGTCCAGCCGTG-3') and "Y-D mutant" (5'-GGAAGCGGGGACGGATGGTGGGAAGAA-3'). MICs of antimicrobials were determined using broth microdilution according to EUCAST methodologies [13].

## Overexpression and purification of recombinant AAC(6')-Ib-cr

The *aac(6′)-Ib-cr* open reading frame was amplified from the synthetic gene template using primers (5'-AAGTTCTGTTTCAGGGCCCGAGCAACGCAAAAACAAAGTTAG-3' and 5'-ATGGTCTAGAAAGCTTTAGGCATCACTGCGTGTTC-3') and subcloned into the T7 expression vector pOPINF [45] using the InFusion recombinase system (Takara Bio) according to manufacturer's instructions. The integrity of the resulting construct was confirmed by sequencing (Eurofins Genomics). The recombinant *aac(6')-Ib-cr*:pOPINF vector was used to transform the *E. coli* T7 expression strain SOLUBL21 (AMS Biotechnology), a single colony was picked and grown overnight in LB growth medium and the resulting culture used for 1% inoculation of 8 x 500 mL of 2 x YT in 2L conical flasks grown at 37˚C with 180 rpm shaking. When absorbance at 600 nm reached 0.6, expression was induced with 0.5 mM IPTG, the temperature was reduced to 18˚C and the culture was grown for a further 18 h. Cells were harvested by centrifugation (Beckman-Coulter Avanti J-26XP, 6500 x g), the supernatant was discarded and pellet resuspended in buffer A (50 mM Tris pH 7.8, 150 mM NaCl, 10 mM Imidazole). Subsequent steps were carried out at 4˚C. The cell suspension was homogenised and sonicated (Sonics Vibra-Cell VC-505, Sonics & Materials Inc., Newton, CT, USA), 1 s

pulse mode, 64% intensity) before ultracentrifugation at 100,000 x g (Beckman Coulter Optima L-80 XP) for 45 min to remove insoluble material. The supernatant was applied to a 5 mL HisTrap Ni-NTA column (Cytiva Life Sciences) pre-equilibrated in buffer A, the column was washed until absorbance returned to base line and bound proteins eluted in buffers B (50 mM Tris pH 7.8, 150 mM NaCl, 20 mM imidazole) and C (50 mM Tris pH 7.8, 150 mM NaCl, 500 mM imidazole). Fractions adjudged, on the basis of inspection of SDS-PAGE gels, to contain AAC(6')-Ib-cr were pooled, concentrated by centrifugal filtration (Sartorius) and loaded onto a Superdex-S75 column (Cytiva Life Sciences) pre-equilibrated with buffer D (50 mM Tris pH 7.8, 150 mM NaCl, 5 mM dithiothreitol (DTT)). Fractions containing AAC(6')-Ib-cr were identified from SDS-PAGE gels and pooled.

## Measurement of enzyme activity

Acetylation of aminoglycoside antibiotics was monitored using the chromogenic 5,5′-dithio-bis-(2-nitrobenzoic acid) (DTNB) assay to detect the free thiol generated on donation of an acetyl group from acetyl-CoA. Reaction rates were measured in UV-star half area clear 96-well microplates (Greiner Bio-One) using a BMG Labtech Clariostar microplate reader. Assays were carried out in triplicate, with a final volume of 100 μl/well, in 50 mM MES pH 6.1 supplemented with 500 μg/mL Bovine Serum Albumin, 2 mM DTNB, 0.1 mM acetyl-CoA, 100 nM purified AAC(6')-Ib-cr and varying concentrations of the aminoglycoside substrates kanamycin B or amikacin. Enzyme activities were calculated by monitoring absorbance at 412 nm over time using a molar absorption coefficient of $13,600 \text{ M}^{-1}.\text{cm}^{-1}$.

## Determination of transcriptional start site using 5'-rapid amplification of cDNA End (5' RACE)

An overnight culture of an *E. coli* isolate carrying the IS*26*-*aac(6')-Ib-cr*-*bla*$_{OXA-1}$-IS*26* transposon was transferred to a flask containing 30 mL of LB broth, adjusted to a starting OD$_{600}$ of 0.1. The flask was incubated with shaking (160 rpm) at 37˚C until reaching an OD$_{600}$ of 0.7. Two millilitres of culture were centrifuged at 4000 rpm for 10 min. Total RNA was extracted from the pellet using Quick-RNA Fungal Bacterial MiniPrep (Zymo Research) following the manufacturer's protocols. Extracted RNA was quantified using Qubit RNA HS Assay Kit (Invitrogen) and Qubit 4 Fluorometer (Thermo Fisher Scientific). The *aac(6')-Ib-cr* transcriptional start site was determined through using the 5' RACE System (Invitrogen) according to manufacturer's instructions. Briefly, first strand cDNA synthesis was performed with a total of 1 μg of RNA and 2.5 ρmol of gene-specific primer1 (GSP1; 5'-TGGTCTATTCCGCGTACTCC-3'). The mixture was incubated at 37˚C for 10 min, then chilled on ice for 1 min before the addition of components at a final concentrations: 1x PCR Buffer, 2.5 mM MgCl$_2$, 10mM DTT, 400 μM dNTP, 200 units SuperScript II to a total volume of 25 μl. The reaction was then incubated at 42˚C for 50 min and terminated by incubation at 70˚C for 15 min. Removal of RNA template was done by adding 1 μl of RNase mix to the mixture and incubated 37˚C for 30 min. First strand cDNA product was purified by passing through a S.N.A.P column according to the manufacturer's procedure. cDNA was eluted in 50 μl of nuclease free water.

Homopolymeric dC-tailing of cDNA was performed in a total volume of 25 μl, using 10 μl of S.N.A.P-purified cDNA sample mixed with 1x tailing buffer containing 200 μM (final) dCTP. The mixture was incubated 94˚C for 3 min, then chilled on ice for 1 min before adding 1 μl TdT and incubating at 37˚C for 10 min. The tailing reaction was heat inactivated by incubating at 65˚C for 10 min.

PCR amplification of dC-tailed cDNA of *aac(6')-Ib-cr* was performed directly by using 5 μl of dC-tailed cDNA from the preceding step as DNA template. A total PCR reaction comprised

DNA template mixed with 25 μl of 2x REDTaq ReadyMix PCR Reaction Mix (Sigma Aldrich), 20 ρmol of Abridged Anchor Primer and GSP2 primer (5'-TTTCTTCTTCCCACCGTCCG-3') and nuclease-free water to a total volume of 50 μl. The amplification cycling was initiated with a 5 min incubation at 95˚C before 35 cycles of denaturation at 95˚C for 30 s, annealing at 56˚C for 30 s and extension at 72˚C for 30 s and final extension at 72˚C for 5 min. PCR products were separated and visualised using 1% agarose gel (Bioline) containing 0.1 μg/ml ethidium bromide. The PCR product was excised from the gel, then gel purified using QIAquick Gel Extraction Kit (Qiagen) according to manufacturer's instructions. The purified PCR product was sequenced using Sanger sequencing with GSP-2 primer.

### Proteomic analysis

A volume of 1 mL of overnight liquid culture was transferred to 50 mL of fresh Cation Adjusted Muller Hinton Broth and incubated at 37˚C until an $OD_{600}$ of 0.5 to 0.6. Samples were centrifuged at 4,000 rpm for 10 min at 4˚C and the supernatants discarded. Cells were resuspended into 35 mL of 30 mM Tris-HCl, pH 8 and broken by sonication using a cycle of 1 s on, 1 s off for 3 min at amplitude of 63% using a Sonics Vibra-Cell VC-505. Non-lysed cells were pelleted by centrifugation at 8,000 rpm (Sorvall RC5B Plus using an SS-34 rotor) for 15 min at 4˚C. The supernatant was used as a direct source of protein for analysis. LC-MS/MS was performed and data analysed as described previously [46,47] using 5 μg of protein for each run. Analysis was performed in triplicate, each from a separate batch of cells. Abundance for each test protein in a sample was normalized relative to the average abundance of ribosomal proteins in the same sample.

### Supporting information

**S1 Table. Contains details of STs, antibiotic susceptibility and ABR genes carried by 3GC-R *E. coli* from bloodstream infections, plus 3GC-S *E. coli* carrying *aac(6')-Ib-cr* sequenced for this study.**
(DOCX)

### Acknowledgments

We are grateful to laboratory staff at Severn Pathology, North Bristol NHS Trust, for collecting the bloodstream isolates, and to Dr Jacqueline Findlay, formerly of the School of Cellular & Molecular Medicine, University of Bristol, for preparing the 2018 isolates for sequencing. WGS was performed by MicrobesNG, Birmingham, UK. We are indebted to Dr Tara deBoer and particularly to Dr Nicole Jackson, BioAmp Diagnostics for provision of isolate-specific susceptibility testing data essential for our secondary analysis of their publicly available genome sequence data.

### Author Contributions

**Conceptualization:** Frances Edwards, Fergus Hamilton, James Spencer, Philip B. Williams, Matthew B. Avison.

**Data curation:** Matthew B. Avison.

**Formal analysis:** Punyawee Dulyayangkul, Thomas Beavis, Winnie W. Y. Lee, Robbie Ardagh, Frances Edwards, Ian Head, Kate J. Heesom, Marek Murarik, Peechanika Pinweha, Carlos Reding, Naphat Satapoomin, John M. Shaw, Yuiko Takebayashi, Philip B. Williams, Matthew B. Avison.

**Funding acquisition:** Kate J. Heesom, Philip B. Williams, Matthew B. Avison.

**Investigation:** Punyawee Dulyayangkul, Thomas Beavis, Winnie W. Y. Lee, Robbie Ardagh, Frances Edwards, Ian Head, Kate J. Heesom, Oliver Mounsey, Marek Murarik, Peechanika Pinweha, Naphat Satapoomin, John M. Shaw, Yuiko Takebayashi, Catherine L. Tooke.

**Methodology:** Carlos Reding.

**Software:** Carlos Reding.

**Supervision:** Oliver Mounsey, Catherine L. Tooke, James Spencer, Philip B. Williams, Matthew B. Avison.

**Visualization:** Matthew B. Avison.

**Writing – original draft:** Punyawee Dulyayangkul, Thomas Beavis, Matthew B. Avison.

**Writing – review & editing:** Punyawee Dulyayangkul, Thomas Beavis, Winnie W. Y. Lee, Robbie Ardagh, Frances Edwards, Fergus Hamilton, Ian Head, Kate J. Heesom, Oliver Mounsey, Marek Murarik, Peechanika Pinweha, Carlos Reding, Naphat Satapoomin, John M. Shaw, Yuiko Takebayashi, Catherine L. Tooke, James Spencer, Philip B. Williams, Matthew B. Avison.

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
