## [Decision Letter · Decision Letter 0]

20 Feb 2024

Dear Professor Avison,

Thank you very much for submitting your manuscript "Harvesting and amplifying gene cassettes confers cross-resistance to critically important antibiotics" for consideration at PLOS Pathogens. As with all papers reviewed by the journal, your manuscript was reviewed by members of the editorial board and by several independent reviewers. In light of the reviews (below this email), we would like to invite the resubmission of a significantly-revised version that takes into account the reviewers' comments.

Dear Dr. Avison,

Thank you for submitting your manuscript to PLoS Pathogens. I have now received three reviews. All three Reviewers noted the importance of your findings but expressed concerns that require your attention. The items of concern listed in the critiques will require results from additional experiments or editorial changes. Most critically, you need to provide direct experimental evidence on promoter utilization (see comments 1 and 3 of Reviewer #1). On this point, if you are unable to do so it would be possible to transfer your paper to PLoS One provided that you are agreeable.

We cannot make any decision about publication until we have seen the revised manuscript and your response to the reviewers' comments. Your revised manuscript is also likely to be sent to reviewers for further evaluation.

Sincerely,

William M Shafer, Ph.D.

Guest Editor

PLOS Pathogens

Matthew Wolfgang

Section Editor

PLOS Pathogens

Michael Malim

Editor-in-Chief

PLOS Pathogens

orcid.org/0000-0002-7699-2064

Dear Dr. Avison,

Thank you for submitting your manuscript to PLoS Pathogens. I have now received three reviews. All three Reviewers noted the importance of your findings but expressed concerns that require your attention. The items of concern listed in the critiques will require results from additional experiments or editorial changes. Most critically, you need to provide direct experimental evidence on promoter utilization (see comments 1 and 3 of Reviewer #1). On this point, if you are unable to do so it would be possible to transfer your paper to PLoS One provided that you are agreeable.

Reviewer's Responses to Questions

**Part I - Summary**

Reviewer #1: This manuscript describes the characterization of an integron-base gene cassette that is present as a transposon flanked by IS26 elements. The work is clearly presented, but is missing key experiments. In addition, there are no page or line numbers in the manuscript which makes it difficult to point out changes. All page references below will be based on the PDF available to this reviewer.

Reviewer #2: The authors offer a succinct and informative research piece detailing the mechanisms underlying pip/taz and amikacin resistance in Enterobacterales carrying AAC(6')-Ib-cr and OXA-1. While the identification of the IS6 transposition carrying these genes may not be novel, the demonstration of IS6-mediated increased expression and amplification of the transposon presents novel findings, specifically, in the context of amikacin and pip/taz resistance. I enjoyed reading this study.

Reviewer #3: *** I have never peer reviewed using the part 1, 2 and 3 model. The intention of grouping into 2 and 3 was to suggest areas where the suggested change is stronger than stylistic but not to inflate the number of type 2 comments to cause a rejection ***

Carbapenem sparing options for therapy remain important notwithstanding the introduction of carbapenemase-inhibitors. This is especially true in resource limited settings and in countries without any marketed carbapenemase-inhibitors.

The major challenge with using PIPERACILLIN-TAZOBACTAM as such a strategy comes from the MERINO trial finding of "not non inferior". There are several reasons why that result may have occurred, and one of them is the inability of conventional testing to have properly inferred PIP-TAZO susceptibility (particularly in OXA-1).

On that background, the idea of using amikacin non-susceptibility to infer the potential presence of OXA-1 is intriguing and the methods used have shown the mechanism of why this should be considered and investigated.

I do worry a bit about some residual misclassification of amikacin and pip-tazo susceptibility because I believe that the Vitek cards (and EUCAST disc diffusion) can make errors, particularly when the organism is CEFTRIAXONE resistant and the "pretest probability" of resistance is higher. I think this needs to be acknowledged somewhere as a limitation. I don't think it invalidates the work -- after all, clinical labs are using these methods today and clinicians are acting upon them.

**Part II – Major Issues: Key Experiments Required for Acceptance**

Reviewer #1: 1. Pg. 11: has the location of the aac(6)-Ib-cr promoter been experimentally verified? If so, this should be referenced here. If not, how do the authors know that the deletion encompassed this region or that no mutations were in the promoter region in amikacin sensitive isolates.

2. Pg 12: Is the statement “In all sequenced aac(6’)-Ib-cr-positive BSI E. coli in our survey, a copy of the insertion sequence IS26 was found upstream of aac(6’)-Ib-cr” referring to amikacin resistant isolates? If so, please clarify.

3. The existence of a hybrid promoter conferred by the IS26 should be experimentally verified by 5’-RACE or primer extension analysis. Without this information, the single bp change in the proposed -10 region may not even be in a promoter region and may be altering gene expression in a post-transcriptional manner. In addition, the effects of the single base substitution on aac transcript levels should be experimentally verified by qRT-PCR analysis.

4. The authors proposed that an IS mediated duplication increased copy number of the aac gene. What were the endpoints of the amplified region? This information should be provided. Also, the specific copy number of genes within the amplified region should be determined by qPCR analysis.

Reviewer #2: I find no major issues with this work

Reviewer #3: Introduction:

I can tell where you are trying to take the first section of the introduction -- and I agree, that losing AG, ceftriaxone, and pip-tazo potentially leaves us little choice (especially since quinolone resistance is also frequently found in ARO with CTX-M). I just think this could benefit from some further reworking from the clinical context if the argument being advanced is clinical.

Results:

“We took this as evidence that aac(6’)-Ibcr is not always a determinant of amikacin resistance.” This assumes that the phenotypic test for susceptibility was correct (and that the breakpoints are correct). To cross reference with methods.

*** NB in 2023 CLSI has updated Amikacin to be a breakpoint of <=4 for susceptibility based on achievable drug levels in humans vs. toxicity. I wonder if/when EUCAST will follow.

And 4 may not be aggressive enough where predicted target attainment is as poor as 20% at 4. See:

https://academic.oup.com/ofid/article/7/7/ofaa084/5804675

This doesn’t invalidate what you are presenting, but would it change your 20% susceptible? Would it also change the “specificity” of the phenotype of amikacin R predicting OXA-1 if Amikacin S is limited to 4 instead of 8. Worth exploring?

Discussion:

Reference 35 is not a randomized controlled trial -- it is mislabelled in pubmed. It is a retrospective cohort study.

Methods

The challenge with pip-tazo disc diffusion by EUCAST is that it can misclassify (see Henderson's MERINO paper in CID comparing to BMD). Would be worth adding this limitation (recognizing that most clinical labs do NOT do BMD).

How well do vitek MICs correlate with MICs from BMD? The tables should indicate you are using Vitek MICs. I am unsure if the Vitek has been validated for the 2023 CLSI amikacin breakpoints. Which vitek card(s) were used? Different cards may have different operating characteristics. Did that card have pip-tazo? If yes, why use disc as truth vs vitek? I agree with the decision but it should be justified.

Table 1 - please make clear the method of MIC.

**Part III – Minor Issues: Editorial and Data Presentation Modifications**

Reviewer #1: 1. Introduction, page 8: E. coli should be Escherichia coli for the initial description.

2. Introduction, page 9 top: exists should be exist.

3. Fig. 3 Legend: More detail should be provided. For example, what does “mutant promoter” indicate?

Reviewer #2: in paragraph one of the introduction, I think that the last sentence could be rewitten to remove "etc" from the list.

the authors conducted a blastn screening of the transposon against genomes available on NCBI, revealing its presence in 714 genomes. Three questions arise from this screening: (1) Was the screen confined solely to the Enterobacterales? (2) Were specific thresholds (%ID, %COV, score) applied to confirm the presence or absence of the transposon? (3) Did the blast screen include only "complete" genomes, or were draft genomes also considered? If the latter, were the criteria for confirming presence/absence consistent across draft and complete genomes?

Regarding the assertion that the transposon is "widely and globally demonstrated," can the authors confirm whether the 714 carriers of the transposon indeed represent a globally distributed collection? Have they ensured that this collection excludes repeated strains from outbreaks or single individuals? It is recognized that such data may not be universally available.

For cases where carriage is confirmed, have the authors delved into the Sequence Read Archive (SRA) for associated sequence read data that might provide insights into copy number?

In the discussion, the authors highlight an important experiment that could enhance the manuscript, although it may not be deemed a major issue. Exploring copy number and its correlation with exposure to amikacin would be intriguing. Specifically, investigating the number of copies required for a transition from susceptible to non-susceptible could provide valuable insights.

Please ensure that all WGS data is deposited to appropriate database as soon as possible.

Reviewer #3: Introduction

Technically, only about 30% of patients with sepsis will have a BSI. Maybe use “frequently” instead and consider combining the first two setences.

I feel like ARO is the common acronym for antibiotic resistant organisms (rather than ABR). Any way you can rework? Not essential.

In many jurisdictions, “the antibiotics of choice tend”

[realistically the empiric antibiotics used in a centre will be a function of the risk/proportion of ARO at that centre or in that patient population]

Extended spectrum beta-lactam resistance is most commonly mediated by CTX-M…. I would think TEM is more common overall (ampicillin/cefazolin)

When people can’t use a beta-lactam, they could use a carbapenem before an aminoglycoside [e.g., in North America] except in certain European settings.

Results:

The (imperfect but important) relationship between amikacin and oxa-1 is well articulated and interesting. Is this pip-tazo resistance independent of being ceftriaxone resistant [e.g.you appear to be describing a system within the CTX-M-15 subgroup -- but if CTX-M-15 isn’t there, is there enough OXA-1 to still be piperacillin-tazobactam R like is seen with the TEM hyperproducers?]

Discussion:

The issue with MERINO was that susceptibility testing relied on an E-test which was recalled for being inaccurate. BMD testing showed that a fair number of isolates were, actually, pip-tazo resistant (particularly as breakpoints have been changed) and that the presumed mechanism for this resistance was OXA-1 co-production.

I agree that what you are proposing regarding using amikacin susceptibility to “screen” for the potential for OXA-1 could be interesting as a carbapenem sparing strategy but that it would need validation in other contexts.

Figures:

Not essential at all -- just curious:

What would figures 4 panels C and D look like as a function of MIC? Worth exploring?

(Figure 5 can’t be done because you don’t have MICs, right?) -- what about as a function of zone size?

PLOS authors have the option to publish the peer review history of their article (what does this mean?). If published, this will include your full peer review and any attached files.

Reviewer #1: No

Reviewer #2: No

Reviewer #3: No

Figure Files:

While revising your submission, please upload your figure files to the Preflight Analysis and Conversion Engine (PACE) digital diagnostic tool, <a href="https://pacev2.apexcovantage.com/">https://pacev2.apexcovantage.com</a

---

## [Editor Report · Decision Letter 1]

30 Apr 2024

Dear Dr. Avison,

We are pleased to inform you that your manuscript 'Harvesting and amplifying gene cassettes confers cross-resistance to critically important antibiotics' has been provisionally accepted for publication in PLOS Pathogens.

Best regards,

William M Shafer, Ph.D.

Guest Editor

PLOS Pathogens

Matthew Wolfgang

Section Editor

PLOS Pathogens

Michael Malim

Editor-in-Chief

PLOS Pathogens

orcid.org/0000-0002-7699-2064
---

## [Editor Report · Acceptance letter]

14 May 2024

Dear Professor Avison,

We are delighted to inform you that your manuscript, "Harvesting and amplifying gene cassettes confers cross-resistance to critically important antibiotics," has been formally accepted for publication in PLOS Pathogens.

Best regards,

Michael Malim

Editor-in-Chief

PLOS Pathogens

orcid.org/0000-0002-7699-2064